# Blows or Falls? Distinction by Random Forest Classification

**DOI:** 10.3390/biology12020206

**Published:** 2023-01-29

**Authors:** Mélanie Henriques, Vincent Bonhomme, Eugénia Cunha, Pascal Adalian

**Affiliations:** 1Centre for Functional Ecology (CEF), Laboratory of Forensic Anthropology, Department of Life Sciences, University of Coimbra, 3000-456 Coimbra, Portugal; 2Aix Marseille Univ, CNRS, EFS, ADES, 13007 Marseille, France; 3Athéna, Lacamp, 30440 Roquedur, France; 4National Institute of Legal Medicine and Forensic Sciences, 3000-456 Coimbra, Portugal

**Keywords:** forensic science, blunt force trauma, falls, blows, skeletal fractures, CT scan, random forests

## Abstract

**Simple Summary:**

In forensic anthropology, skeletal trauma analysis can assist pathologists in determining the circumstance, cause, and manner of death. Determining whether the trauma is related to falls or induced by homicidal blows is often asked in relevance to legal issues. The hat brim line rule (HBL) is one of the most commonly used methods. The rule says that fractures resulting from blows may be found above and within the HBL, not on the skull’s base. Recent studies have found that the HBL rule must be used carefully, and postcranial skeletal trauma could be useful in this distinction. Evidence presented in court must follow Daubert’s guidelines for validity and reliability (evidence validated; error rates known; standards available; findings should be peer-reviewed and accepted by the scientific community). In this study, we assessed skeletal fracture patterns resulting from both etiologies. We tested various models for the method; the best one was based on the binary coding of 12 anatomical regions or 28 bones with or without baseline (age and sex). The results show the possible identification of the etiology in 83% of the cases. This method could be helpful for forensic experts in the interpretation of bone fractures.

**Abstract:**

In this study, we propose a classification method between falls and blows using random forests. In total, 400 anonymized patients presenting with fractures from falls or blows aged between 20 and 49 years old were used. There were 549 types of fractures for 57 bones and 12 anatomical regions observed. We first tested various models according to the sensibility of random forest parameters and their effects on model accuracies. The best model was based on the binary coding of 12 anatomical regions or 28 bones with or without baseline (age and sex). Our method achieved the highest accuracy rate of 83% in the distinction between falls and blows. Our findings pave the way for applications to help forensic experts and archaeologists.

## 1. Introduction

Considerable research has been conducted to attempt to distinguish between falls and blows. The most cited approach is the hat brim line (HBL) rule, which has often been simplified as fractures resulting from falls located within the HBL. In contrast, fractures resulting from blows may be found above and within the HBL, not on the skull’s base. Recent studies have found that the HBL rule must be used very carefully because their results do not match the definition [1,2,3,4].

We previously showed that the discrimination between falls and blows could be discussed by the site and the number of fractures found on the skull and the trunk [4].

This study primarily aimed to find additional valuable criteria (i.e., the type of fractures) in the distinction of both etiologies. Furthermore, we aimed to test various models with random forests by selecting and combining criteria with the highest predictability rates.

## 2. Materials and Methods

### 2.1. Dataset Description

Following the standards of the National Consultative Ethics Committee for health and life sciences (CCNE), the National Council of Ethics for the Life Sciences (CNECV), and the Helsinki Declaration of 1975 concerning the privacy and confidentiality of personal data, our dataset consisted of 400 anonymized patients presenting with fractures from falls or blows and between 20 and 49 years old. The CT scans of our sample were collected from the PACS (Picture Archiving and Communication System) in the Assistance Publique-Hôpitaux de Marseille (AP-HM, Marseille, France), the Centro Hospitalar e Universitário de Coimbra, and the Centre Hospitalier Regional et Universitaire de Nancy. According to the acquisition protocol, the scanner slices were 0.6 mm and 1.25 mm thick. Each individual was reviewed in the three anatomical planes (axial, coronal, and sagittal) using the window viewing presets for bone and adjusted manually on AW Workstation (AW server 2.0, GE HealthCare, Milwaukee, WI, USA) and Horos^®^ (version 3.3.5, ^©^ 2021 Horos Project). Three-dimensional volume renderings were also used to identify the fractures.

The following variables were available: sex and age (later referred to as baseline) on the one hand and the 549 types of fractures for all 57 bones on the other. We used two classifications, AO/OTA and Galloway, Wedel 2014 (Broken Bones), when possible. Otherwise, we observed the presence/absence of a fracture in different parts of a bone [5,6]. The observations were performed using multiplanar reconstructions (MPRs), maximum intensity projection (MIP), and volume rendering (VR) reconstructions on Horos version 3.3.5^®^.

To cope with absent and rare events (less than 5%), such as fractured bones with low frequencies, we excluded 534 types and 29 bones. The final dataset included 15 types of fractures and 28 bones.

On the 28 remaining bones, 12 anatomical regions were defined: cranium, basicranium, cranial vault, face, mandible, scapula, ribs, thoracic vertebrae, lumbar vertebrae, sacrum, coxal bone, and femur.

### 2.2. Inter- and Intra-Observer Errors

To assess the repeatability, we randomly selected 30 individuals from the sample. Inter- and intra-observer variations were evaluated using Cohen’s kappa coefficient with the {KappaGUI} R package.

### 2.3. Random Forest Approach

We aimed to predict the circumstances of observed fractures, i.e., a blow or a fall, using the available etiology. We chose to use the random forest approach because it is an appropriate supervised learning technique when the number of observations is lower or of the same magnitude as the number of variables [7,8]. It is also adapted to classification problems which include qualitative and quantitative variables.

### 2.4. Statistical Environment

All analyses were performed in the R 4.1.3 statistical environment [9], using the following packages: randomForest to model and predict using random forests [10]; pROC to calculate ROC curves [11]; and tidyverse for general data manipulation, programming, and data visualization [12].

### 2.5. Model Selection

As implemented in randomForest [10], the random forest algorithm comes with three internal parameters: mtry (the number of variables randomly sampled as candidates at each split), nodesize (the minimum size of terminal nodes), and ntree (the number of trees to calculate). Additionally, our dataset enabled different approaches: which data to use (bones, typology, anatomical region, and whether or not to include the sex/age of the patient) and the metric of the data (quantitative, ternary, or semi-qualitative {0, 1, 2+} or binary {0/1}).

Altogether, these five parameterizing dimensions enable many different models to be trained. The successive steps described below aim to reduce this number to a few accurate models.

### 2.6. Grid Search for Hyper Parameters Optimization

We first explored the sensibility of random forest parameters and their effects on model accuracies. We used a grid search approach on the five dimensions. The mtry parameter was the only one that varied between datasets. A default and sensible value for these parameters is the square root of the number of variables used, rounded to the lower integer. We circumvented this by defining mtry_k as simply a multiplicative factor to this default value. For example, for a 36-variable dataset, mtry_k = 1 leads to mtry = 6, mtry_k = 0.5 results in mtry = 3, mtry_k = 2 produces mtry = 12, etc.

The full combination of models tested was: mtry_k {0.25, 0.5, 1, 2, 4}; nodesize {1, 2, 5, 10}; ntree {101, 501, 1001, 2001}; metric {quantitative, ternary, binary}; datasets {baseline alone, bone, bone + baseline, type, type + baseline, anatomical region, anatomical region + baseline}. This resulted in 1680 models.

The dataset was randomly partitioned into 300 patients for the training set and 100 patients for the testing set. The latter was never seen by the model while training. To estimate parameter elasticity and the impact of such partitioning, we repeated the entire grid search process for ten different sets of partitions, following the same scheme.

### 2.7. Benchmarking Models with Fixed Internal Parameters

After selecting random forest internal parameters, we ran the same models and explored the structure of their predictions, including the contrast between the error obtained on the train versus on the test partition.

Model selection was also performed here to select both the dataset and metric to use. Accuracy, i.e., low error, was the first criterion. We also considered how the models were generalized: ideally, we would expect similar errors to indicate that the model was not overfitting training data. Finally, parsimony helped us select between metric: for comparable model performance, the more straightforward (e.g., binary versus ternary), the better.

### 2.8. Class-Wise Predictions for the Final Models

Finally, on the four final models, we explored their results as regards their prediction in terms of etiology alone, etiology for each sex, and etiology for age classes. To ease graphical interpretation, we binned the continuous age variables into 10-year bins ranging from 20 to 49.

### 2.9. Variable Importance and Their Sign

Variable importance, i.e., how the bone/anatomical results influence the classification task, was calculated. We also attempted signing these contributions towards either blow or fall. This could not be retrieved directly with random forests; however, the marginal distributions of occurrence for each bone/region enabled accurate estimates to be performed. The proportions of broken bones/regions were calculated and adjusted for the overall sample size of each etiology; otherwise, it was unbalanced.

### 2.10. Predicting New Patients

Regarding new individuals, there is no guarantee of complete information. Some fractures may not be recorded, and some bones may just be missing, making it difficult to assess whether they were broken or not when the person passed away. In forensic contexts, there are many cases where information about the context is unknown and where it is paramount to establish the manner of death, i.e., whether it was accidental, homicide, or suicide.

## 3. Results

### 3.1. Inter- and Intra-Observer Errors

The inter- and intra-observer errors were evaluated using Cohen’s kappa (Table 1) [13,14]. A table taken from Landis and Koch [15] was used for agreeing to the evaluation (Table 2).

The results showed a perfect and substantial agreement for most variables with a binary quotation. Only the eighth rib exhibited a moderate inter-observer error, with a kappa value of 0.44. Only fractures in the petrous portion of the temporal bone have a sight error inter-observer (0).

The results showed a perfect and substantial agreement for most variables with a ternary quotation. Only the third, eighth, and tenth ribs showed moderate inter-observer error, with kappa values of 0.5, 0.48, and 0.54, respectively. Only fractures in the petrous portion of the temporal bone exhibited a slight inter-observer error (0).

The results show a perfect and substantial agreement for most variables with a quantitative quotation. Other variables showed a moderate inter-observer error, between 0.48 and 0.60, for fractures in the basicranium, ribs (third, eighth, and tenth), and sacrum. Other variables showed a moderate intra-observer error between 0.54 and 0.57 for ribs and thoracic vertebrae fractures.

The best reproducibility was in all types of quotation of the type of fractures, the ternary quotation of fractures in anatomical regions, and, more generally, the binary quotation.

### 3.2. Parameter Optimization

As shown in Figure 1, we ran each model several times for different values of the mtry_k, nodesize, and ntree parameters. For each run, we measured the model’s error rate (i.e., 1-accuracy).

For each box figure, the green box indicates that the central 50% of data lies in this section; the bold bar is the median value; the upper and lower black bars are the greatest and least values, excluding outliers; and finally, the black crosses represent the outliers.

As seen from the box figure, all rating models had similar thresholds for the error rate despite the variations in the mtry_k, nodesize, and ntree parameters.

However, notably, the mtry_k = 1, nodesise = 5, and ntree = 501 parameters showed a lower error rate than models based on the observation of fractures in the bone with or without baseline and anatomical regions.

### 3.3. Benchmarking Models

Various models were tested using the parameters selected above (Figure 2).

The error of the models based on the type of fractures or only with baseline was 35% or greater.

The best models were inferior to 25% and were based on the bone or the region of fractures, with or without the baseline. Thus, for the next step, we omitted the type of fractures.

The results were similar for the three rating systems (binary, ternary, and quantitative). We decided to keep the binary quotation because it showed very few inter- and intra-observer errors.

### 3.4. Exploring Class-Wise Predictions for the Final Four Models

The results of the final four models, bone, bone and baseline, region of fracture, and region of fracture and baseline, are presented in Figure 3.

The second part of the figure shows the relationship between context and sex. The rate error in falls for females varied between 0% and 20%; for males, it was between 14% and 28%. For blows, the rate error varied for females between 17% and 40%; for males, it was between 12% and 23%.

The third part of the figure shows the relationship between context and age. The rate error in falls for individuals aged between 20 and 29 years was 17% to 30%; for individuals aged between 30 and 39 years old it was 12% to 20%; and for individuals aged between 40 and 49 years old, it was 10% to 32%. The rate error in blows for individuals aged between 20 and 29 years old was 7% to 15%; for individuals aged between 30 and 39 years old it was 9% to 37%; and for individuals aged between 40 and 49 years old, it was 25% to 61%.

### 3.5. Variable Importance in Model

The variable importance of the final four models, bone, bone and baseline, region of fracture, and region of fracture and baseline, are presented in Figure 4.

The most important predictors in the model based on bone were fractures in the mandible, maxilla, coxal, sacrum, and nasal bone.

The most important predictors in the model based on bone and baseline were fractures in the mandible, the individual’s age, and fractures in the maxilla bone, coxal bone, and sacrum.

The most important predictors in the model based on the anatomical region were fractures in the mandible, cranium, face, lumbar, and ribs.

The most important predictors in the model based on the anatomical region and baseline were the individual’s age and fractures in the mandible, cranium, face, and lumbar.

Fractures in the cranium, mandible on the pelvic girdle, lumbar, and ribs were very important in the distinction between falls and blows.

Notably, the two most essential parameters in these four models were the mandible and the patient’s age.

To observe which localization of fracture tended to be more caused by blows or falls, we created Figure 5.

This figure shows the distribution of the most relevant parameters in the distinction between falls and blows.

The left shows the fractures in bones. Fractures in the mandible, maxillary, nasal, zygomatic, and ethmoid bones are more frequent due to blows than falls. No fractures of the fifth lumbar vertebrae, sacrum, and femur were observed in blow cases. The other fractures were more present in fall cases than in blow cases.

The right side of Figure 5 shows the classification of fractures present in the anatomical regions. Fractures in the mandible, face, and cranium were more frequently observed as a result of blows than falls. No fractures in the sacrum and femur were observed in blow cases. Fractures in the basicranium, vault, lumbar vertebrae, ribs, thoracic vertebrae, scapula, and coxal bone were more frequent in fall cases than in blows.

## 4. Discussion

### 4.1. Repeatability

The results presented a substantial to perfect agreement for most of the variables, especially in the case of binary quotation.

### 4.2. Fracture Location, Sex, and Age

Figure 3 shows differences between the rate of error between males and females and between the age classes in blow cases. The error rate was important for females and individuals aged between 40 and 49 years old.

These differences could be explained by the fact that the context of fractures in medical reports may be misinformed or because of the bone’s quality. Bone can be weakened by pregnancy or lactation, or in postmenopausal females, among other things [16,17,18,19]. This fragility would be more conducive to fractures.

### 4.3. Model of Prediction

Random forests enabled us to construct models optimized on the observed data, determining new classification criteria. The best prediction models were based on a binary quotation of fractures in 12 anatomical regions or 28 bones with or without baseline (age and sex). These models enabled correct predictions between 77% and 83%.

Fractures in the basicranium, vault, lumbar vertebrae, ribs, thoracic vertebrae, scapula, and coxal bone were more common in fall cases than in blows. These results are concordant with those of Henriques et al. [4].

We discuss fractures in the basicranium and the cranial vault further because this is a particular subject in distinguishing between blows and falls.

According to the literature, thoracolumbar injuries can result from motor vehicle accidents, falls from a significant height, and direct blows [16,20,21,22,23,24,25,26].

Rib fractures are common injuries from sports, direct blows or kicking, falls, high-velocity trauma, and cardiopulmonary resuscitation [27,28,29,30,31,32,33,34].

Pelvic fractures are common in motor vehicle accidents, falls, and sport-related accidents [6,16,35,36].

According to the literature, scapula fractures can result from falls, motor vehicle incidents, or direct blows [6,16,36,37,38].

In our study, fractures in the mandible, maxillary, nasal, zygomatic, and ethmoid bones are more frequent due to blows than falls. Most of these results are concordant with those of Henriques et al. [4].

This is relevant to the study by Wulkan et al. [39] on interpersonal violence-caused panfacial fractures. As for isolated bone structures, the mandible and the nasal had the highest incidence of fractures. Panfacial fractures involve fractures of the frontal bone, maxilla, zygomatic complex, nasoethmoid-orbital region, sphenoid, and mandible [40]. In our case, frontal and sphenoid bone fractures were more frequent in falls than blows.

According to Laski et al. [41], the most frequent etiology of facial trauma is assault (75%), and mandible fractures occur in 46.7% of cases.

The head and the neck are most commonly affected by violence [42].

When we approach the subject of the distinction between blows and falls based on skull fractures, it is difficult not to think of the hat brim line (HBL) rule created by Walcher in 1931 [43]. This refers to an area of the skull between two lines parallel to the Frankfurt horizontal plane; the superior line passes through the glabella, and the inferior line runs through the external auditory meatus [1].

This rule has often been simplified as fractures resulting from falls are often located within the HBL, whereas fractures resulting from blows may be found above and within the HBL, but not on the skull base [1,2,43,44,45].

Some studies have shown that it is more complex, even if the idea that fractures above the HBL occur in cases of falls remains recurrent.

Ehrlich and Maxeiner observed that lacerations from blows more often occur above the HBL than from falls [46,47].

Kremer et al. [1,2] showed that injuries from blows are more often found above the HBL; a laceration within the HBL is more in favor of a fall; and a skull fracture within the HBL is found equitably in both etiologies.

The results from Guyomarc’h et al. [44] showed that blows can be distinguished from falls due to four criteria, including fractures above the HBL.

Our results showed that fractures in the vault and the basicranium occurred more frequently in fall cases than in blows. Our previous study demonstrated the same results [4]. This is discordant with the HBL rule.

According to the HBL rule, fractures resulting from blows will not be found on the skull base.

This last point can also be contradicted by the study by Ta’ala et al. (2006); we should focus on the context of trauma, because their research revealed that cranial trauma was more likely caused by execution with a variety of blunt weapons applied to the back of the head/neck by Khmer Rouge soldiers [48]. Moreover, a victim can fall during an assault.

Rogers [49] wrote that fractures in the basicranium could indirectly result from blows to the front of the head or through the compression of the spine against the base of the skull.

Research by Lefèvre et al. (2015), regarding differences in injuries caused by falls from less than 2.5 m high and homicides, the incidence of cranial fractures in both etiologies was similar [50]. In their study, the HBL could have been more helpful in distinguishing between falls and blows.

The difference in the occurrence of fractures between this study and ours can be explained by the different heights of falls. Lefevre et al. (2015) selected low falls; we did not select a height for falls [50]. Greater heights result in a wider distribution and greater severity of fractures than accidental falls [34,37].

Many studies have shown that fractures and injuries on the cranial vault and above the HBL could result from repeated falls, falls from a height, or an impact against an edge or a corner, such as falls involving stairs [1,2,3,43,43,44,46,47,51].

According to Geserick et al. (2014), the HBL rule does not apply to blows and falls from a height (including from stairs) [3].

The HBL rule suggests that fractures from falls do not lie above the hat brim line when some conditions are fulfilled (i.e., standing position of the individual before falling, flat floor without incline or stairs, falling from one’s height, or the absence of intermediate obstacles) [3,43].

These application parameters are often omitted, which could explain the results of studies based on the HBL rule.

Additionally, the HBL rule should be used with caution because studies of the discrimination of falls and blows are based only on fractures in the skull (cranial vault and basicranium).

The localization of fractures in the cranium is not discriminatory of one etiology or another; however, according to our results, the presence of fractures in the fifth lumbar vertebrae, sacrum, and proximal extremity of the femur seems to be for blow cases.

According to the literature, thoracolumbar injuries occur in falls from a significant height [16,20,21,22,23,24,25]. However, fractures of the lumbar transverse processes may occur due to direct blows to the lumbar area [26].

The research performed by Mullingan and Talmi on 357 cases of assault found two patients with lumbar spine transverse process fractures at the L5 level, but not on the pelvis or the femur [52].

Many researchers have found that femur fractures result from falls from heights [6,16,36,53].

Sacral fractures frequently occur in falls. However, these can also occur due to direct blows [54,55,56,57,58,59]. Some cases, such as that studied by Berryman and Saul, present cases of violent sexual assault with a fractured sacrum caused by a tire iron inserted vaginally [60].

Our method can be helpful to forensic experts in determining the manner and death and the distinction between homicidal death by blunt trauma and falls.

Moreover, anthropologists and pathologists can testify in courtrooms; evidence presented in court must follow Daubert’s guidelines for validity and reliability (evidence validated; the error rates known; standards available; and the findings should be peer-reviewed and accepted by the scientific community) [61,62].

To the best of our knowledge, no method of distinction between falls and blows has been proposed based on a statistically viable sample, with error rates known and strong statistics.

Our results show that it is possible to discuss the etiology when determining the probability of belonging to one etiology or another.

However, notably, this method was based on individuals aged between 20 and 49 years old. Its application and results are not assured for individuals not belonging to the same age group.

Moreover, this method will be tested on a forensic sample; for easy use and response, it will be developed in applications to register fractures.

## 5. Conclusions

In this study, we investigated fall and blow distinctions using random forest classification. The results indicated a good separation between the two etiologies using binary coding on 12 anatomical regions or 28 bones. Further evaluation of this classification system is needed as validation on forensic subjects with the mode of occurrence of fractures known.

However, these preliminary results support the possibility that this system for distinguishing falls from blows could be a relevant tool for experts, with prediction rates between 77% and 83%.

In future studies, we intend to develop an application to register fractures and give us the probability that these come from one etiology or another.

## Figures and Tables

**Figure 1 biology-12-00206-f001:**
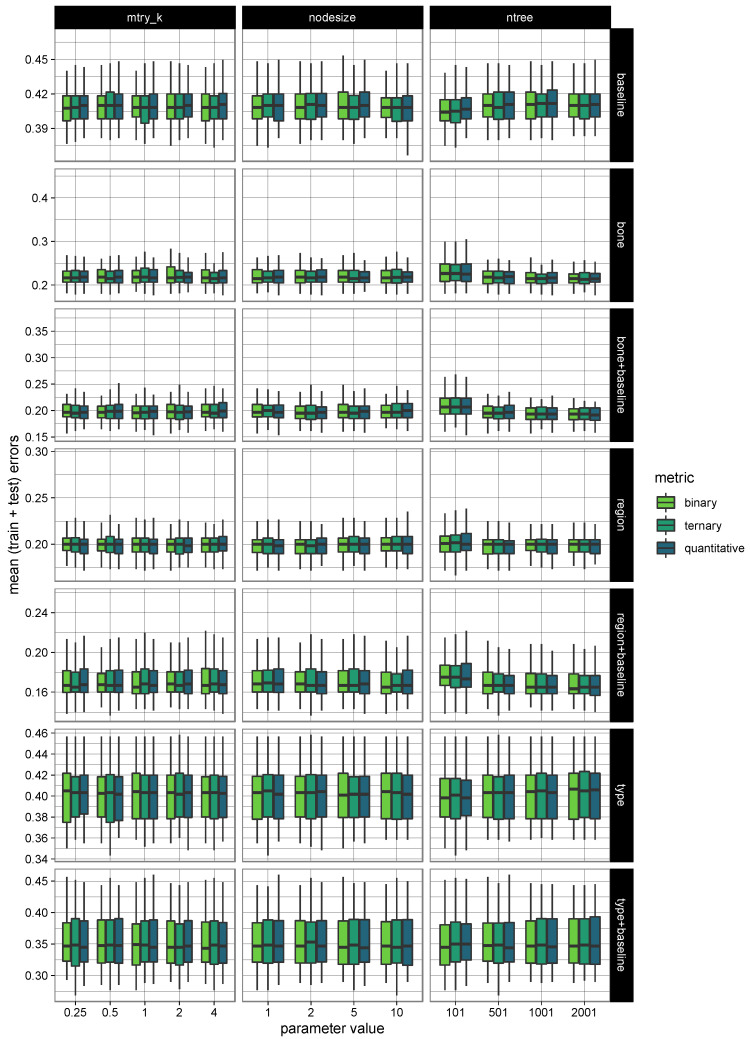
Random forest parameter optimization of all the models.

**Figure 2 biology-12-00206-f002:**
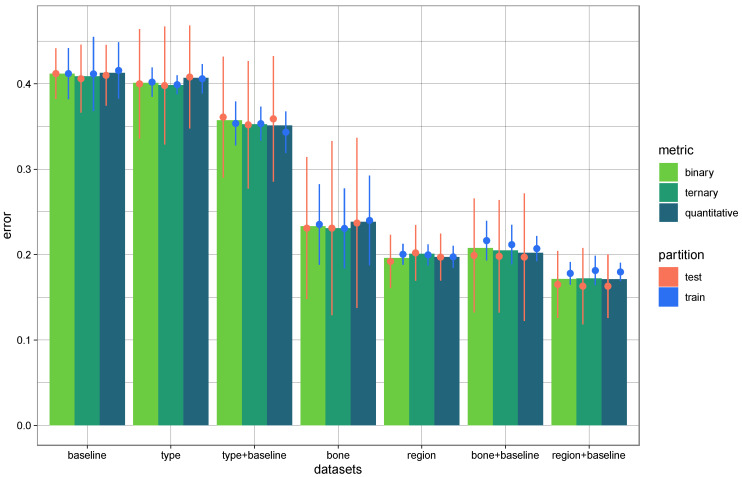
Error rates of all the models in test and train samples by random forest.

**Figure 3 biology-12-00206-f003:**
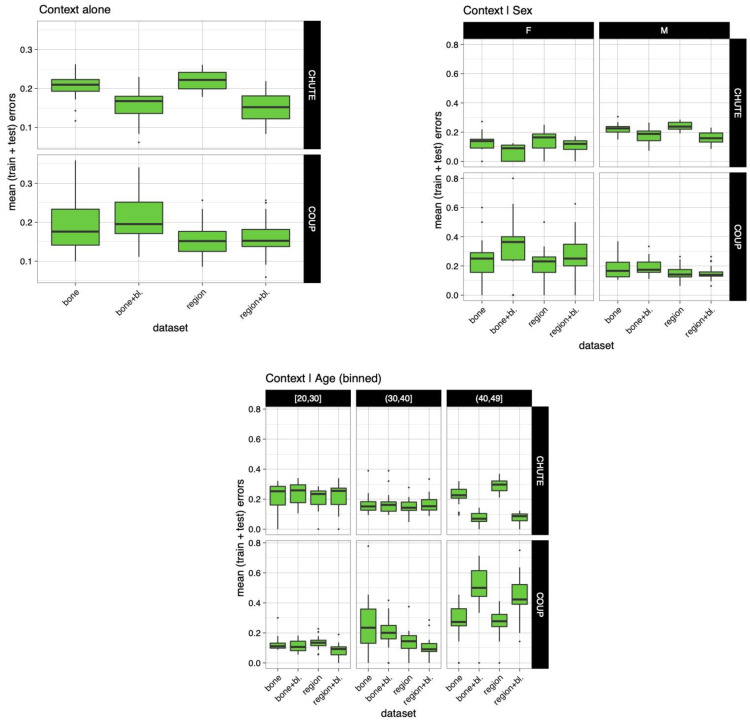
The error rate of the models was based on fractures in the anatomical regions and bone with or without baseline by etiology, sex, and age.

**Figure 4 biology-12-00206-f004:**
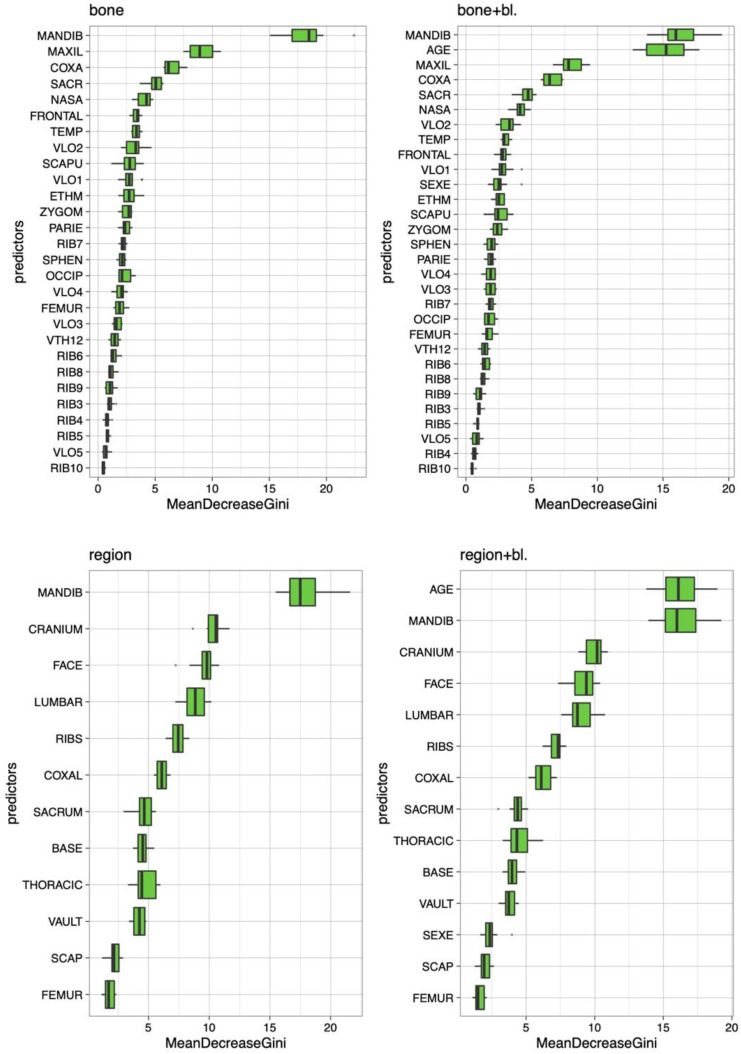
Variable importance for the four final models based on fractures in the anatomical regions and bones with or without baseline.

**Figure 5 biology-12-00206-f005:**
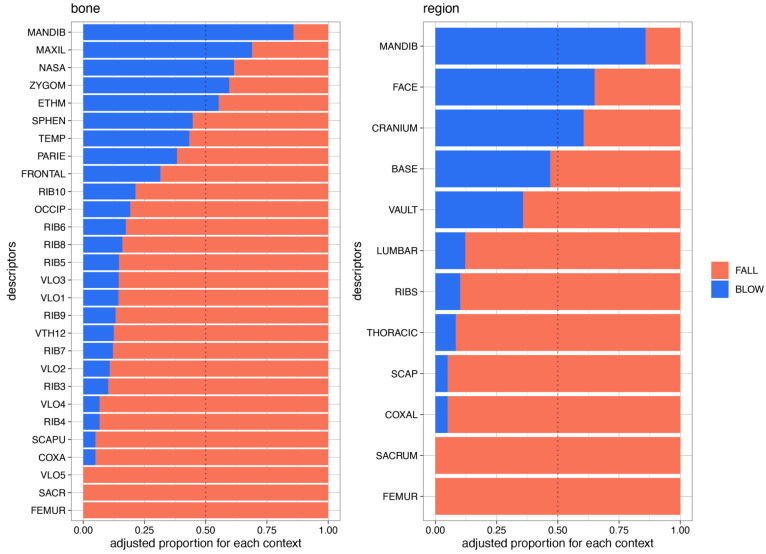
Distribution of the fractures in the anatomical region and bones according to the etiology.

**Table 1 biology-12-00206-t001:** Inter- and intra-observer errors in the assessment of the presence and the number of fractures in fourteen anatomical regions using Cohen’s kappa.

Localization	Absence/Presence		Absence/Simple/Multiple		Quantitative	
	Inter-Observer	Intra-Observer	Inter-Observer	Intra-Observer	Inter-Observer	Intra-Observer
Basicranium	0.71	1	0.72	1	0.60	0.78
Cranial Vault	0.84	1	0.84	1	0.84	1
Face	0.9	0.9	0.91	0.82	0.82	0.82
Mandible	0.87	1	0.75	1	0.75	1
Scapula	0.84	1	0.84	1	0.84	1
Ribs	0.72	1	0.68	0.93	0.41	0.57
Thoracic V.	0.76	0.84	0.77	0.68	0.78	0.54
Lumbar V.	0.92	1	0.92	0.93	0.71	0.75
Sacrum	1	1	0.86	1	0.59	0.86
Coxal	1	0.87	1	0.87	1	1
Femur	1	1	1	1	1	1
Frontal	0.78	1	0.79	1	0.79	1
Parietal	1	1	1	1	1	1
Occipital	1	1	1	1	1	1
Temporal	0.84	1	0.84	1	0.69	1
Sphenoid	0.84	1	0.84	0.72	0.84	0.72
Ethmoid	1	1	1	1	1	1
Nasal	1	1	1	1	1	1
Maxilla	0.71	0.84	0.72	0.84	0.72	0.84
Zygomatic	1	1	0.86	0.86	0.86	0.86
Mandible	0.87	1	0.75	1	0.75	1
Scapula	0.84	1	0.84	1	0.84	1
Rib3	0.61	1	0.5	1	0.50	1
Rib4	0.76	1	0.77	1	0.77	1
Rib5	1	1	1	1	0.86	1
Rib6	0.84	0.87	0.84	0.88	0.69	0.88
Rib7	0.67	0.81	0.68	0.83	0.68	0.83
Rib8	0.44	0.91	0.48	0.84	0.48	0.84
Rib9	0.90	0.90	0.82	0.91	0.73	0.82
Rib10	0.76	0.89	0.54	0.89	0.54	0.89
VTH12	0.64	1	0.64	1	0.64	1
VLO1	1	1	0.90	0.90	0.81	0.81
VLO2	1	1	1	0.90	1	0.90
VLO3	0.71	1	0.71	1	0.71	1
VLO4	0.84	0.87	0.84	0.87	0.84	0.87
VLO5	0.71	0.76	0.71	0.76	0.71	0.76
Coxal	1	0.87	1	0.87	1	0.87
Sacrum	0.84	1	0.84	1	0.84	1
Femur	1	1	1	1	1	1
Simple fracture zygomatic process of Temporal	0.78	1	0.78	1	1	0.78
Petrous portion of Temporal	0	1	0	1	0	1
Linear fracture of Sphenoid	1	0.78	1	0.78	1	1
Body of Sphenoid	1	1	1	1	1	1
Ethmoid	1	1	1	1	1	1
Nasal	1	1	1	1	1	1
Simple Fracture Maxilla	0.65	0.65	0.65	0.65	0.65	0.65
Comminuted fracture Maxilla	0.78	1	0.78	1	0.78	1
Ascending ramus of Mandible	1	1	1	1	1	1
Comminuted fracture of Mandible	0.65	1	0.65	1	0.65	1
Transverse process of VL1	1	1	1	1	1	1
Transverse process of VL2	1	1	1	1	1	1
Transverse process of VL3	0.63	1	0.63	1	0.63	1
Transverse process of VL4	0.78	0.84	0.78	0.84	0.78	0.84
Comminuted fracture of Coxal bone	1	1	1	1	1	1

**Table 2 biology-12-00206-t002:** Cohen’s kappa agreement (Data from Landis and Koch (1977)).

Kappa (k)	Strength of Agreement
<0	Disagreement
0.00–0.20	Insignificant
0.21–0.40	Low
0.41–0.60	Middle
0.61–0.80	Good
0.81–1.00	Very good

## Data Availability

The data for this study are kept by the first author.

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
