# Peer review of "Blows or Falls? Distinction by Random Forest Classification"

_biology, 2023, doi:10.3390/biology12020206_

Round 1

Reviewer 1 Report

Useful analysis. thank you to the authors. I have some concern relating to the bounds of the age range of the data set (max 49 years).Authors findings for analysis regarding 40-49 year set need qualifying by mentioning that findings relating to that age group is restricted by the  data set and it is unclear whether the same analysis and findings apply to individuals older than 49 or not.
In places in the paper the language is shorthand and needs expressing more carefully and in complete English sentences.

Author Response

Thank you for your kind and insightful comments, which helped us to clarify some points and thus improve the overall quality of our submission.

Indeed, our results are certainly not applicable to another age group than that of our study. We mentioned this when correcting the manuscript.
As for the English, we will have it corrected by the MDPI.

Best regards

Reviewer 2 Report

Thank you for the opportunity to review your manuscript for Biology -"Blows or falls? Distinction by Random Forests Classification"The premise of the paper is interesting and the idea of applying Random Forests Classification to distinguish blows and falls can give us new informations.

The writing is generally good and clear and the work is well organized and methodologically sound.

The analysis is appropriate, but your figures (3 and 4 on page 8 and figure 9 on page 5, together with the descriptions written below) are small and difficult to read. You might consider moving to a landscape orientation and using the full size of the page for these graphs.

I suggest you to properly underline one of the key point of your scientific contribution which is the comparison between the classic knowledge around HBL and your results, which is buried in the text.

With some basic revision, this will be an useful scientific contribution.
Thank you for your work on this issue.

Author Response

Thank you for your kind and insightful comments, which helped us to clarify some points and thus improve the overall quality of our submission.

We enlarged the figures and improved their quality.

We reworked the discussion regarding our scientific contribution in comparison to HBL.

As for the English, we will have it corrected by the MDPI.

Best regards

Reviewer 3 Report

The manuscript is interesting, from the point of separation between blows and falls and from the statistical data manipulation. However, it needs to be improved. I put notes directly in the manuscript.

General notes:

Some grammar mistakes that need to be corrected. Some sentences are also not well written, either they are missing something or are structured wrongly. I marked with yellow (without a note) just some of them as English is not my first language. Nevertheless, I think the manuscript needs proofreading and corrections of mistakes. Figure captions also need to be corrected and some Figures need better resolution. 

 Also, terminology is hard to follow as sometimes the same thing is addressed differently. Occasionally I was lost. For example, I am not sure what class represent.

The structure is quite fractographic, which is hard to read especially in the part about statistics and discussion. For the latter, it seems as they are just stating results from the previous researchers without meaningful incorporation of their results. In general, discussion needs some work. I think it needs better correlation between the obtained results and included results of other researchers.

 In the conclusions, they could elaborate a bit more on putting the results into some more tangible context.

Author Response

Thank you for your kind and insightful comments, which helped us to clarify some points and thus improve the overall quality of our submission.

We didn’t see your notes in the manuscript, if there was anything other than simple summary, author contributions, funding, institutional Review Board Statement, Informed Consent Statement, Data Availability Statement, Conflicts of Interest.

We enlarged the figures and improved their quality.

We reworked the discussion regarding our scientific contribution in comparison to other studies and putting on context.

"Class" represent group.

As for the English, we will have it corrected by the MDPI.

Best regards